# Duckweed: Beyond an Efficient Plant Model System

**DOI:** 10.3390/biom14060628

**Published:** 2024-05-27

**Authors:** Doni Thingujam, Karolina M. Pajerowska-Mukhtar, M. Shahid Mukhtar

**Affiliations:** 1Department of Biology, University of Alabama at Birmingham, 3100 East Science Hall, 902 14th Street South, Birmingham, AL 35294, USA; dthinguj@uab.edu; 2Department of Biological Sciences, Clemson University, 132 Long Hall, Clemson, SC 29634, USA; 3Department of Genetics & Biochemistry, Clemson University, 105 Collings St. Biosystems Research Complex, Clemson, SC 29634, USA

**Keywords:** duckweed, plant model, phytoremediation, sustainability, transgenic

## Abstract

Duckweed (*Lemnaceae*) rises as a crucial model system due to its unique characteristics and wide-ranging utility. The significance of physiological research and phytoremediation highlights the intricate potential of duckweed in the current era of plant biology. Special attention to duckweed has been brought due to its distinctive features of nutrient uptake, ion transport dynamics, detoxification, intricate signaling, and stress tolerance. In addition, duckweed can alleviate environmental pollutants and enhance sustainability by participating in bioremediation processes and wastewater treatment. Furthermore, insights into the genomic complexity of *Lemnaceae* species and the flourishing field of transgenic development highlight the opportunities for genetic manipulation and biotechnological innovations. Novel methods for the germplasm conservation of duckweed can be adopted to preserve genetic diversity for future research endeavors and breeding programs. This review centers around prospects in duckweed research promoting interdisciplinary collaborations and technological advancements to drive its full potential as a model organism.

## 1. Introduction

Duckweed, also known as *Lemnaceae*, is a rapidly proliferating aquatic monocot angiosperm. The common habitat of *Lemnaceae* constitutes still or slow-moving freshwater environments, including ponds, lakes, and marshes. *Lemnaceae* species are very tiny with a size ranging from 1 to 10 mm in length consisting of floating leaves called fronds and delicate root forms referred to as rootlets. With a relatively small structure, *Lemnaceae* species can double the biomass with a high nutritional value within a short period under favorable growth conditions [1,2]. Duckweed can serve as a model organism for understanding physiological aspects relative to growth and development due to its simple morphology, rapid growth, and ease of cultivation. Moreover, duckweed possesses unique adaptive characteristics that enable it to survive in diverse environmental conditions, including low-light conditions, fluctuating nutrient levels, and different ranges of pH [3]. Furthermore, duckweed has an efficient potential to remove heavy metals, pesticides, and organic compounds from wastewater sources. Therefore, duckweeds have gained attention as a promising candidate for investigating plant stress responses and environmental interactions due to their distinct ability to accumulate a variety of pollutants and to survive in nutrient-rich and polluted waters, their high protein content, and their potential as a source of biofuel [4]. Although the *Lemnaceae* family is extensively diverse and consists of more than 30 species with five genera, *Lemna*, *Landoltia*, *Spirodela*, *Wolffia*, and *Wolfiella*, most research has focused on the *Lemna* genus in bioaccumulation and tolerance to environmental stressor studies [5]. Additionally, the genome size of the *Lemnaceae* species is relatively small compared to the other monocots which allow duckweed to further contribute to genomic and transcriptomic studies [6]. The understanding of genomics and transcriptomics in duckweed can enable the biotechnological assessment of genetic modifications and their impact on innovative sustainable solutions for environmental and agricultural challenges.

## 2. Advantages of Duckweed over *Arabidopsis*

*Arabidopsis thaliana* (*Arabidopsis*), a small cruciferous dicot plant, is a universally accepted model plant for plant biology research [7]. *Arabidopsis* offers many advantages to be used as a functional and classical model for studying various aspects of plant research, from physiology to multi-omics. It has a high reproductive potential and due to its small size, it can be grown on a large scale in a very small space. Additionally, it can be a self-pollinated plant, and thus it is easier to regulate the extraneous variables during experiments in comparison to cross-pollinated plants [8]. Also, the completion of the entire genome sequencing of *Arabidopsis* by the *Arabidopsis* Genome Initiative in the year 2000 makes it a popular experimental model [9]. Although *Arabidopsis* exhibits ample advantages, it has certain limitations in understanding the aspects of crop plants, particularly the monocots. Recently, duckweed has re-emerged as a compelling alternative or complement to the *Arabidopsis* for the monocot model system. The remarkable growth rate and nutrient acquisition of duckweed highlight its significant advantage as a model organism [10]. Rapid growth enables quicker experiments and high-throughput screening, especially for genetics and large-scale studies. Subsequently, it acquires a small genome size, typically around 150 Mbp, which simplifies genomic analysis and genetic manipulation [6]. In addition, duckweed offers distinct advantages over *Arabidopsis*, including adaptability to various environmental conditions, and potential for bioenergy production [11]. Duckweed can be an excellent model organism for biological studies, including genetics, genomics, ecology, and renewable energy research. Considering the several advantages associated with duckweed, it can be used for different research studies (Figure 1).

## 3. Importance of Duckweed in Physiological Research

### 3.1. Nutrient Uptake and Stress Tolerance

With relatively minimal size and remarkable growth and adaptations, duckweed has become the interest of plant physiological studies. The focus on duckweed has been expanded due to its efficient and rapid nutrient absorption from the surrounding ecosystem with an extensive growth rate. Like terrestrial plants, duckweed can uptake nutrients including nitrogen and phosphorus that are essential for growth and metabolism through passive diffusion, active transport, and symbiotic interactions with microorganisms. However, the nutrient uptake in duckweed is a non-continuous process and is solely affected by environmental abundance [12]. A study evaluated the effectiveness of *L. minor* in removing nutrients from both synthetic and dumpsite leachates under artificial and natural climate conditions. The assessment involved cultivating duckweed in controlled conditions and open conditions using leachate derived from decomposed solid waste collected from various residential, commercial, and industrial dumpsites and synthetic ones. To prepare the dumpsite leachate, the collected waste was processed and filtered, while a synthetic leachate mimicking the composition of dumpsite leachate was also prepared. Duckweed was grown on both types of leachates over a 10-day period in June and July, with controlled pH levels maintained throughout the experiments. The study measured parameters such as chemical oxygen demand (COD), total phosphorus (TP), ammonium nitrogen (NH_4_^+^-N), orthophosphate (o-PO_4_-3-P), total Kjeldahl nitrogen (TKN), and duckweed biomass at the beginning and end of the test periods. The results from this study indicated that duckweed efficiently removed nutrients and COD from dumpsite leachate under controlled conditions. However, nutrient absorption, specifically phosphate and nitrogen, was approximately 35% and 16% higher in synthetic leachate. Additionally, duckweed growth was significantly accelerated in the synthetic leachate environment. These outcomes suggested that *L. minor* is highly effective for nutrient removal, especially under controlled conditions, although the growth and nutrient absorption rates vary depending on the type of leachate [12]. However, the regulatory mechanisms governing nutrient uptake processes in duckweed are not well understood. The research approach with particular attention on physiological and developmental processes can provide a scope with an enormous advantage in nutrient cycling and sustainable agriculture. Additionally, duckweed has significant photosynthetic capabilities with high rates of carbon fixation and efficient light energy utilization [11]. An essential aspect of understanding the photosynthetic system is the light adaptation potential. Several physiological studies have shown the effect of different light intensities on the growth and adaptation of diverse *Lemnaceae* species. With high-intensity light, *L. gibba* species reduce the physiological growth rate alternatively, diverting the growth to defense regulation with the increased production of protective pigments such as carotenoids showing their strategic adjustments to light conditions [13]. In addition, prior investigations have elucidated the response of duckweed to environmental stressors encompassing heavy metal exposure, salinity variations, and temperature oscillations [14]. In *L. minor* species, the increased accumulation of macro- and micronutrients exhibited the proficiencies of duckweed to overcome salinity-induced stress [15]. The ability to tolerate environmental stressors is another promising characteristic of duckweed that provides insight into plant stress resistance responses.

### 3.2. Ion Transport in Duckweed

Duckweed has made substantial contributions to understanding the aspect of physiological ion transport. In a groundbreaking study, the dynamic response of calcium (Ca^2+^) signaling in duckweed under cadmium (Cd) stress conditions was elucidated through the utilization of a Ca^2+^-sensing fluorescent reporter (GCaMP3) in transgenic duckweed (*Lemna turionifera 5511*). The notable accumulation of Ca^2+^ in vacuoles revealed the subcellular calcium localization during Cd stress. Furthermore, the Ca^2+^ inflow study suggested that Ca^2+^ inflow was stable at a slow speed; however, the treatment of Cd changed to high-speed efflux. Additionally, the introduction of exogenous γ-aminobutyric acid (GABA) to duckweed resulted in the stabilization of the Ca^2+^ signal, highlighting its significant regulatory function in managing the Ca^2+^ signal under Cd-induced stress [16]. In addition to Cd, cobalt (Co^2+^) is another metal ion that has a phyto-modulatory effect in duckweed species. The hyperaccumulation of Co^2+^ in fronds inhibits the vegetative growth of *L. minor*, with no distinctive alterations in the iron (Fe) content in fronds. The chlorophyll content and photosynthetic efficiency initially remained constant but gradually decreased over time with the acquisition of Co^2+^, suggesting the inhibition of biosynthesis rather than promoting the degradation of existing pigment molecules. This finding indicates the strength of *L. minor* as a model system for understanding the mechanistic effect of heavy metal metabolism and bioaccumulation at the cellular level [17].

Apart from heavy metals, the transport and impact of other ions such as phosphate (P*_i_*), nitrate (NO^3−^), and ammonium (NH_4_^+^) have been investigated in different duckweed species. In the pre-anthropogenic era, P*_i_* was the limiting mineral factor for floating aquatic plants, including duckweed, under natural conditions. Therefore, duckweed and other macrophytes have evolved to be particularly proficient in assimilating and storing this ion. Moreover, P*_i_* deficiency in duckweed has been shown to exert significant impacts on growth and metabolic processes. The induction of a glycosylphosphatidylinositol-anchored purple acid phosphatase (PAP) in *L. punctata* in response to P*_i_* deficiency facilitates enhanced P*_i_* uptake and storage in vacuoles to support growth. Furthermore, duckweeds such as *L. minor* and *L. gibba* employ diverse strategies for P*_i_* storage, including the accumulation of various phosphate forms as short- and long-term P*_i_* reserves [18]. Therefore, duckweed can survive under phosphorus deficiency, highlighting its adaptability and making it an invaluable model for studying the nutrient acquisition and storage mechanisms in aquatic plants.

### 3.3. Signaling Mechanisms in Duckweed

In addition to ion transport, duckweed has been a promising system for studying plant signaling responses to various environmental cues and stressors. Several studies have investigated the signaling pathways involved in the growth, development, and stress tolerance mechanisms of duckweed. Previous findings suggested the roles of phytohormones, including auxins, cytokinins, abscisic acid (ABA), and gibberellins, in regulating key processes such as cell division, differentiation, biomass growth, starch accumulation, and stress responses in diverse duckweed species, including *L. gibba*, *S. polyrhiza*, *L. turionifera*, and *L. punctata* [19,20,21,22,23]. Furthermore, it has been demonstrated that nutrient starvation induces significant metabolic changes in duckweed. A comparative transcriptome evaluation of *L. punctata* under nutrient starvation revealed high starch accumulation. It was found that nutrient deprivation triggers the activity of ADP–glucose pyrophosphorylase, a key starch synthesis enzyme, that reprograms metabolic processes to enhance starch biosynthesis as a survival strategy [24]. Additionally, the involvement of protein kinases, Ca^2+^, and reactive oxygen species (ROS) in mediating signal transduction pathways has been identified in duckweed, particularly in response to environmental stresses such as salinity, oxidative stress, and heavy metal toxicity [25,26,27]. Protein kinases, including MAPKs and CDPKs, are also integral to the stress response mechanism in duckweed. Under salinity stress, it has been revealed that *S. polyrhiza* exhibits distorted ion homeostasis and activates specific signaling pathways involving protein kinases to restore cellular balance and promote stress adaptation. These protein kinases phosphorylate various target proteins, leading to the activation of stress-responsive genes [25]. The involvement of Ca^2+^ in the duckweed response to stress has been highlighted in various studies. For instance, a study on *L. turionifera* demonstrated that Ca^2+^ signaling is crucial for responding to Cd stress, suggesting that both glutamate (Glu) and gamma-aminobutyric acid (GABA) are involved in the stress response mechanism. This complex signaling network demonstrates that Ca^2+^ acts as a key messenger in mitigating Cd toxicity by modulating metabolic pathways. Similarly, ROS, which includes superoxide anion (O_2_^•−^) and hydrogen peroxide (H_2_O_2_), are not only by-products of cellular metabolism but also crucial signaling molecules that activate antioxidant defenses [26]. Another investigation of *L. minor* observed that short-term exposure to Cd leads to oxidative stress, characterized by increased ROS production. The gradual increase in oxidative stress triggers the activation of antioxidant enzymes, including catalase and peroxidases, and other protective mechanisms used to mitigate cellular damage [27]. Likewise, the signaling components, such as receptors, transcription factors, and downstream effectors, involved in developmental processes, responses to pathogens, and environmental stimuli have been characterized through genomics and transcriptomics analysis [28,29,30]. The understanding of duckweed signaling networks can give scope to develop strategies to improve its resilience and productivity under adverse environmental conditions.

## 4. Phytoremediation and Wastewater Treatment through Duckweed

Duckweed has garnered attention for its potential applications in phytoremediation and wastewater treatment due to its ability to efficiently remove nutrients, heavy metals, and organic pollutants from aquatic environments [4]. Research has demonstrated that *S. polyrhiza*, *L. minor*, and *L. punctata* can effectively absorb and accumulate nutrients such as nitrogen and phosphorus, thereby mitigating eutrophication in water bodies [31]. Additionally, duckweed can uptake and sequester heavy metals such as cadmium, lead, and arsenic, contributing to the remediation of contaminated water and soil [14]. Furthermore, duckweed has the potential to degrade organic pollutants through mechanisms such as phytodegradation and rhizofiltration, where pollutants are metabolized by the plant itself or microbial communities associated with its roots. Moreover, duckweed-based constructed wetlands and wastewater treatment systems have been assessed and optimized for the treatment of various types of wastewater, including municipal, agricultural, and industrial effluents [32,33,34]. The disposal of contaminated duckweed biomass remains a significant challenge, particularly when it accumulates heavy metals or other hazardous substances. Following treatment, the contaminated duckweed cannot be disposed of through standard means due to the risk of secondary pollution. Several disposal and utilization strategies can be implemented to address this issue [35]. Thermal treatments such as incineration or pyrolysis are effective methods to safely break down organic materials and immobilize heavy metals in the resulting ash, reducing the risk of environmental contamination, and the ash produced can sometimes be repurposed in construction materials, contributing to waste minimization and resource recovery [35,36]. Another approach is phytomining, where plants are used to extract valuable metals from contaminated biomass, providing economic incentives by recovering metals like zinc, nickel, and cadmium for reuse in industrial processes [35,37,38]. Additionally, composting and anaerobic digestion could be viable options for the disposal of duckweed biomass, provided that the contaminant levels are within safe limits. These processes can convert biomass into biofertilizers or biogas, transforming waste into valuable resources. However, these methods require careful monitoring to ensure that contaminants are adequately managed and do not pose a risk to the environment or human health [39,40]. By implementing these strategies, the environmental risks associated with the disposal of contaminated duckweed can be effectively managed, ensuring that duckweed-based wastewater treatment remains a sustainable and environmentally friendly solution.

## 5. Genome Complexity of *Lemnaceae* Species

Although duckweed has promising characteristics, a few challenges hamper the application of duckweed for agricultural and biotechnological applications. The limited genomics and transcriptomics information of duckweed restricts this powerful model system from flourishing in plant stress research. Moreover, the genomic content of duckweed is found to be enriched with repetitive elements, sequences that act as a confounding factor for sequencing and assembly efforts [41,42,43,44,45]. The major significant challenge is the genomic complexity of duckweeds, including large variations in genome sizes, high levels of repetitive sequences, and polyploidy. The characterization of genomes for duckweeds such as *L. minor*, *S. polyrhiza*, and *Wolffia australiana* revealed high proportions of repetitive elements and transposable sequences, gene duplications, and polyploidy, complicating genome assembly and functional annotation [44,46,47]. These factors make genome assembly and annotation difficult, leading to incomplete or fragmented genomic data [48]. Incomplete genomic information can hinder finding the gene function and regulation. Additionally, the bioinformatics bottleneck, including the need for advanced computational tools and expertise to handle and analyze large datasets, further constrains the effective utilization of genomic and transcriptomic data [49]. These limitations can impact the broader application of findings by hindering the identification of the key genes and pathways involved in important traits such as stress tolerance, growth, development, and signaling processes. Advanced sequencing technologies, such as single-molecule real-time (SMRT) sequencing and nanopore sequencing, offer solutions by providing longer read lengths and higher accuracy, which are crucial for resolving these complex genomic regions [50]. Additionally, comparative genomic studies across different duckweed species can elucidate conserved and unique features, aiding in the development of genetically engineered strains for improved stress tolerance and biomass production [51]. Furthermore, transcriptomic studies face limitations such as variable gene expression levels and the presence of numerous isoforms, which complicate the accurate quantification and characterization of transcripts [52]. To overcome these challenges, several approaches can be employed. Enhancing computational capabilities through parallelization of the assembly process, as observed with tools like ABySS and ALLPATHS-LG, can improve genome assembly and analysis [53,54]. Increasing the processing speed and storage capacity of computers will also facilitate the handling of large datasets. Developing new sequencing platforms that provide longer reads with unbiased coverage can address the issue of complex repeats and improve genome assembly quality. Third-generation sequencing technologies, such as single-molecule sequencing, offer promise with their longer read lengths, and combining these with next-generation sequencing (NGS) can enhance the accuracy and completeness of transcriptomic data [55]. Furthermore, integrating multi-omics approaches, including metabolomics and proteomics, can provide a more comprehensive understanding of plant biology. These techniques can complement genomic and transcriptomic data, offering insights into metabolic pathways and protein interactions that are crucial for developing stress-tolerant and high-yielding crop varieties [56]. In addition to genomics and transcriptomics knowledge, considering the complexity of genomic structure, it is also essential to analyze the genomic conformation of *Lemnaceae*. High-throughput sequencing techniques need to be applied to thoroughly understand its genome conformation. One such sequencing method is PacBio sequencing, which can provide long, detailed reads of the genome, crucial for assembling complex genomes like duckweed [57]. These long reads allow for more complete and accurate assemblies, better gene characterization, and variant detection within the duckweed population. However, PacBio sequencing is not allowed to reveal the 3D organization of the genome. To analyze the 3D architecture of the genome, Hi-C sequencing comes into play to capture the interaction pattern of different regions of chromosomes, providing a blueprint for the physical structure of the genome [58]. This information helps to direct PacBio sequences into their proper chromosomal positions and relative spatial gene interaction potentially influencing stress tolerance [59]. A powerful and inclusive understanding of the duckweed genome can be achieved by combining the detailed sequencing of PacBio with the 3D architecture from Hi-C insights.

## 6. Transgenic Development of Duckweed

Studies on transgenic development in *Lemnaceae* have showcased its potential as a versatile platform for genetic engineering and biotechnological applications. Genetic engineering has been employed to enhance various traits in duckweed, including biomass production, nutrient uptake, stress tolerance, and biofuel production, through different transformation techniques. A pioneering study showed the genetic transformation of duckweed species *L. gibba* and *L. minor* using Agrobacterium-mediated transformation [60]. Another study assessed the feasibility of producing a protective antigen for the PEDV spike protein 1 using duckweed and demonstrated its potential as a bioreactor for producing vaccines. This study involves the stable transformation of *L. minor* through co-cultivation with *Agrobacterium tumefaciens EHA105* harboring the PEDV spike protein gene. The integration and expression of the transgene were confirmed via genomic PCR and RT-PCR, respectively. At the same time, Western blot analysis verified the presence of the PEDV spike protein 1 in the transgenic *L. minor* [61]. Similar to this work, another finding suggested that transgenic duckweed can yield high-quality antigens for the creation of an edible “universal” vaccine against influenza viruses. This study demonstrated that the M2e peptide fused to the ricin B subunit could be expressed in nuclear-transformed *L. minor* without affecting plant morphology or growth rate. Oral immunization with recombinant RTB-M130 protein stimulated an immunological response and produced anti-M2e antibodies in mice. The vaccination assay further exhibited that the ricin B subunit retains its adjuvant capabilities in the fusion protein, increasing the development of anti-M2e antibodies in the vaccinated mice [62]. Furthermore, a rapid and efficient method for agrobacterium-mediated genetic modification allows for the expression of chicken interleukin-17B in *L. minor*, which serves as a mucosal vaccine adjuvant. The plant-produced chIL-17B activated the NF-κB pathway, the JAK-STAT pathway, and their downstream cytokines in DF-1 cells. Additionally, chIL-17B transgenic duckweed was administered orally as an immunoadjuvant with a mucosal vaccine against infectious bronchitis virus (IBV) in chickens. Moreover, the group fed with chIL-17B transgenic plants exhibited significantly higher IBV-specific antibody titers and increased secretory immunoglobulin A (sIgA) concentrations. This approach enhances the immunological applications of duckweed, making it a viable platform for producing medical therapeutics [63]. In the area of biofuel production, *L. japonica* was successfully engineered to accumulate triacylglycerols by modifying lipid biosynthesis pathways. This transformation involved overexpressing genes, an estradiol-inducible CFP-*Arabidopsis* WRINKLED1 fusion protein, a constitutive mouse diacylglycerol:acyl-CoA acyltransferase2 (MmDGAT), and a sesame oleosin variant (SiOLE(*)) that increase the production of fatty acids and their subsequent assembly into triacylglycerols (TAG), crucial components of biodiesel [64]. Furthermore, another finding also shows the potential of duckweed as a platform for expressing cellulolytic enzymes. Specifically, endoglucanase E1 from *Acidothermus cellulolyticus* was successfully expressed in the cytosol of transgenic duckweed (*L. minor* 8627) under the control of the cauliflower mosaic virus 35S promoter. Western blot analysis indicated that the recombinant enzyme co-migrated with the catalytic domain fraction of the native E1 protein, suggesting that the cellulose-binding domain was cleaved near or in the linker region. Importantly, the enzyme expressed in duckweed retained its biological activity, with expression levels reaching up to 0.24% of total soluble protein. The endoglucanase activity, measured using carboxymethylcellulose, averaged 0.2 units per mg of protein extracted from fresh duckweed. These findings demonstrate the feasibility of using duckweed for the production of active cellulolytic enzymes, which could have significant applications in biotechnology, particularly in biofuel production and other industrial processes requiring cellulose degradation [65]. Stress tolerance in duckweed has also been significantly improved through genetic modifications. The overexpression of the *Arabidopsis* gene *serine–glyoxylate aminotransferase* (*AtAGT1*) in *L. minor* led to enhanced salt stress tolerance by optimizing the photorespiratory pathway. This genetic modification helps duckweed maintain cellular function under high salinity conditions [66]. Additionally, the overexpression of the Na+/H+ antiporter (*NHX1*) gene has been shown to reduce cadmium accumulation in *L. turionifera*, providing increased tolerance to heavy metal stress by maintaining ion homeostasis [67]. These genetic transformations, utilizing precise techniques such as Agrobacterium-mediated transformation, rapid gene insertion methods, and targeted gene overexpression, not only improve the productivity and sustainability of duckweed but also expand its applications in biotechnology, environmental management, and medical therapeutics. Furthermore, advances in genome editing technologies, such as CRISPR-Cas9, have enabled precise and targeted modifications of duckweed genomes, accelerating the development of transgenic lines with desired traits [68]. Through these transgenic development studies, duckweed can emerge as a promising platform for sustainable biotechnological applications, including bioenergy production, wastewater treatment, and environmental remediation.

## 7. Development of Novel Methods for Duckweed Germplasm Conservation

The development of new cryopreservation methods for duckweed has aimed to establish efficient techniques for the long-term preservation of genetic diversity and conservation of valuable germplasms. Cryopreservation is crucial for maintaining the genetic resources of duckweed species, especially those with unique traits of interest for biotechnological and agricultural applications. Traditional cryopreservation methods, such as slow freezing and vitrification, have been adapted and optimized for duckweed, but they often suffer from low recovery rates and genetic stability issues [69]. Consequently, it is of utmost importance to explore novel cryopreservation approaches tailored to the unique physiology and morphology of duckweed. One promising technique is encapsulation–dehydration, where plant embryonic tissues are encapsulated in protective calcium-alginate matrices and dehydrated before cryopreservation [70,71]. The calcium-alginate encapsulation and dehydration method can be applied to duckweed for long-term germplasm conservation. Additionally, advancements in cryoprotectant solutions and protocols have been claimed to improve the viability and regrowth of cryopreserved duckweed samples [72]. These innovative cryopreservation methods can offer practical solutions for the long-term storage and conservation of duckweed germplasm, facilitating future research and biotechnological applications. Continued research efforts are needed to optimize protocols, enhance recovery rates, and ensure genetic stability for the widespread adoption of cryopreservation in duckweed conservation programs and biobanking initiatives. Moreover, duckweed has also acquired significant interest as a sustainable food source due to its high nutritional composition [73,74]. Therefore, the preservation of the genetic diversity of duckweed is crucial for plant research and potential food security.

## 8. Discussion and Future Perspectives

Recent developments in duckweed research have advanced its potential applications across different fields, showing enormous potential for this versatile plant in the near future. The traits of rapid growth, high nutrient uptake efficiency, and tolerance to the environmental stresses of duckweed present advantages to exploit it to its full potential. Profound insights into the genetic makeup of duckweed’s unique traits can be studied with the current advancements in genomic sequencing and molecular biology techniques. The elucidation of duckweed genetic makeup can facilitate the development of genetic engineering techniques aimed to enhance the desirable traits in duckweed and other monocot plants. Genetic manipulation can be used to enhance traits such as increased biomass production, nutrient uptake, and stress resistance to maximize the utility of duckweed in various applications. Moreover, innovations in cultivation technologies such as photobioreactors and specialized wastewater treatment techniques might optimize duckweed cultivation for applications in bioenergy production, the bioremediation of polluted water bodies, and sustainable agricultural practices. These technologies can enable the efficient cultivation of duckweed on a large scale for addressing pressing environmental challenges. Furthermore, beyond its traditional applications, duckweeds are being investigated as a promising medium for biopharmaceutical production, the phytoremediation of contaminated sites, and ecological restoration efforts [75,76]. These diverse applications highlight the versatility of duckweed and its potential to contribute to multiple aspects of human well-being and environmental sustainability (Figure 2). Interestingly, duckweed might show promise for boosting space research by providing sustainable solutions for life support and resource management beyond Earth. However, duckweed requires efficient methods for germplasm conservation to utilize their potential to the fullest. The traditional liquid and solid culture media have been used as methods for germplasm preservation and the in vitro maintenance of duckweed plants in the laboratory. Germplasm conservation can be efficiently extended through a novel synthetic seed technology approach. Developing duckweed synthetic seeds (SynSeeds) can be a distinct germplasm conservation method for enabling a consistent supply of duckweed plants for sustainable food production and research. Overall, the future of duckweed research holds promise for further developments in genetic manipulation, cultivation methods, and downstream processing technologies. Continued innovation in these areas is anticipated to establish duckweed as an efficient model system in addition to its adoption as a sustainable solution to global challenges related to food security, environmental degradation, and the transition to renewable energy sources.

## Figures and Tables

**Figure 1 biomolecules-14-00628-f001:**
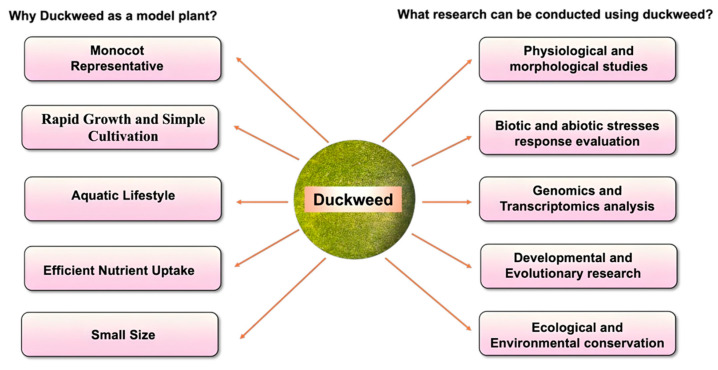
Benefits and utilization of duckweed in diverse research domains.

**Figure 2 biomolecules-14-00628-f002:**
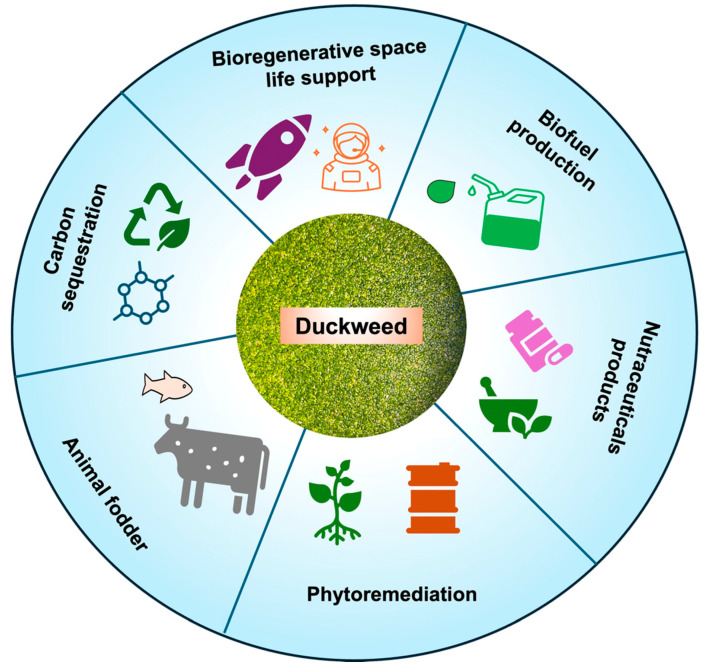
Potential applications of duckweed in promoting environmental sustainability and enhancing agricultural productivity.

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
