# Peer review of "Duckweed: Beyond an Efficient Plant Model System"

_biomolecules, 2024, doi:10.3390/biom14060628_

Round 1

Reviewer 1 Report

Comments and Suggestions for Authors

The manuscript reviews Duckweed's role as a versatile plant model, highlighting its rapid growth, genetic simplicity, and phytoremediation capabilities, alongside its potential in genetic research and environmental sustainability applications. However, there are some problems need to be cared about:

1. The manuscript mentions Duckweed's capability for nutrient uptake and stress response without detailing the methodologies used to study these processes on Lines 80-84. For reproducibility and to strengthen the paper's credibility, it's crucial to include specific experimental designs, conditions, and analytical techniques.

2. On Lines 169-172, the text briefly comments on the challenges related to genomic and transcriptomic studies, but it fails to explore how these limitations could impact the broader application of the findings. Please expanding on these limitations and discussing potential solutions to this.

3. Figure 1 between Line 73-74 is unclear in an enlarged view, and the picture of the duckweed is inappropriate.

4. The manuscript does not provide enough detail on the techniques and their precise impacts on Duckweed's traits on Line 190-197.

5. Several claims about Duckweed's signaling mechanisms and environmental stress responses are made without direct citations to supporting research on Line 150-153, please add the according citations.

Comments on the Quality of English Language

The manuscript reviews Duckweed's role as a versatile plant model, highlighting its rapid growth, genetic simplicity, and phytoremediation capabilities, alongside its potential in genetic research and environmental sustainability applications. However, there are some problems need to be cared about:

1. The manuscript mentions Duckweed's capability for nutrient uptake and stress response without detailing the methodologies used to study these processes on Lines 80-84. For reproducibility and to strengthen the paper's credibility, it's crucial to include specific experimental designs, conditions, and analytical techniques.

2. On Lines 169-172, the text briefly comments on the challenges related to genomic and transcriptomic studies, but it fails to explore how these limitations could impact the broader application of the findings. Please expanding on these limitations and discussing potential solutions to this.

3. Figure 1 between Line 73-74 is unclear in an enlarged view, and the picture of the duckweed is inappropriate.

4. The manuscript does not provide enough detail on the techniques and their precise impacts on Duckweed's traits on Line 190-197.

5. Several claims about Duckweed's signaling mechanisms and environmental stress responses are made without direct citations to supporting research on Line 150-153, please add the according citations.

Author Response

File attached

Reviewer 2 Report

Comments and Suggestions for Authors

- Line 26-32 Reference needed. Suggest        

       Landolt, E. The Family of Lemnaceae—A Monographic Study; Veröffentlichungen des Geobotanischen Institutes der ETH, Stiftung Ruebel: Zurich, Switzerland, 1986; Volume 1.

and 

Landolt, E.; Kandeler, R. The Family of Lemnaceae—A Mono-Graphic Study; Veroeffentlichungen des Geobotanischen Instutes der â€¨ETH; Stiftung Ruebel: Zurich, Switzerland, 1987; Volume 2.

- Lines 45-47 Reference needed. Introduce Reference 8 at this point.

- Lines 164-166 What about the issue of disposal of the contaminated, rapidly growing duckweed plants following treatment of “industrial effluents” (line 163)? Offer solutions, otherwise tone down the rosy rhetoric.

- Lines 214-216  Reference needed.

- Lines 218-220  Reference needed.

- Lines 220-222  Reference needed.

- Lines 229-231  References needed. Introduce Reference 55 at this point. Also, suggest adding at this point 

Edelman, M.; Colt, M. Nutrient value of leaf vs. seed. Front. Chem. 2016, 4, 32.  which provides a broader perspective to duckweed’s “high nutritional composition“.

- Lines 232-238  Either provide a reference or move entirely to the “Discussion and future perspectives” section.

- Lines 250-252  Reference needed for “photo-bioreactors” for “optimized Duckweed cultivation”. Or, change the text to the present tense (“..might optimize Duckweed cultivation for ..”).

- Lines 255-256  Reference needed for  “biopharmaceutical production,” (especially as it appears as an “application” in Figure 2. Neither Ref. 55 or Ref. 56 relate to pharmaceuticals). I suggest adding 

Rival, S., et al. (2008). Spirodela (duckweed) as an alternative production system for pharmaceuticals: A case study, aprotinin. Transgenic Research, 17(4), 503-513.  which is a direct, primary source for biopharmaceutical production in duckweed.

-Lines 261-262  I recommend completely deleting “Appendix A” as it is totally unreferenced. It’s mainly hand waving and detracts from the factual aspects of the Review. In this regard, in Line 260 change  “..Duckweed shows tremendous promise..” to “..might show promise..”

Author Response

File attached

Round 2

Reviewer 1 Report

Comments and Suggestions for Authors

The authors have revised the paper to address my problem. I have no further comments.

Comments on the Quality of English Language

The authors have revised the paper to address my problem. I have no further comments.